# Intended versus actual delivery location and factors associated with change in delivery location among pregnant women in Southern Province, Zambia: a prespecified secondary observational analysis of the ZamCAT

Hiwote Solomon ,[1] Elizabeth G Henry,[2,3] Julie Herlihy,[4] Kojo Yeboah-Antwi ,[5] Godfrey Biemba,[3,6] Kebby Musokotwane,[7] Afsah Bhutta,[8] Davidson H Hamer ,[3,9] Katherine E A Semrau [10,11,12]

DHH and KEAS are joint senior authors.

For numbered affiliations see end of article.

**Correspondence to**
Dr Hiwote Solomon;
solomohi@bu.edu

## ABSTRACT

**Objectives** This prespecified, secondary analysis of the Zambia Chlorhexidine Application Trial (ZamCAT) aimed to determine the proportion of women who did not deliver where they intended, to understand the underlying reasons for the discordance between planned and actual delivery locations; and to assess sociodemographic characteristics associated with concordance of intention and practice.

**Design** Prespecified, secondary analysis from randomised controlled trial.

**Setting** Recruitment occurred in 90 primary health facilities (HFs) with follow-up in the community in Southern Province, Zambia.

**Participants** Between 15 February 2011 and 30 January 2013, 39 679 pregnant women enrolled in ZamCAT.

**Secondary outcome measures** The location where mothers gave birth (home vs HF) was compared with their planned delivery location.

**Results** When interviewed antepartum, 92% of respondents intended to deliver at an HF, 6.1% at home and 1.2% had no plan. However, of those who intended to deliver at an HF, 61% did; of those who intended to deliver at home, only 4% did; and of those who intended to deliver at home, 2% delivered instead at an HF. Among women who delivered at home, women who were aged 25–34 and ≥35 years were more likely to deliver where they intended than women aged 20–24 years (adjusted OR (aOR)=1.31, 95% CI=1.11 to 1.50 and aOR=1.32, 95% CI=1.12 to 1.57, respectively). Women who delivered at HFs had greater odds of delivering where they intended if they received any primary schooling (aOR=1.34, 95% CI=1.09 to 1.72) or more than a primary school education (aOR=1.54, 95% CI=1.17 to 2.02), were literate (aOR=1.33, 95% CI=1.119 to 1.58), and were not in the lowest quintile of the wealth index.

**Conclusion** Discrepancies between intended and actual delivery locations highlight the need to go beyond the development of birth plans and exposure to birth planning messaging. More research is required to address barriers to achieving intentions of a facility-based childbirth.

## Strengths and limitations of this study

► In this large, cluster randomised controlled trial, we were able to gain information on >39 000 pregnant women enrolled with less than 5% loss to follow-up, providing robust estimates of factors related to birth location intentions and actual practice.

► The study was further strengthened by variation in intended location of birth as well as differences in actual practice of delivery location.

► Social desirability bias among respondents as questions such as birth plan and reasons behind delivery choice may have influenced women to respond that they intended to deliver in a facility.

► Qualitative data would be helpful to explore the underlying motivations and decision-making processes around intention and actual delivery locations.

**Trial registration number** ClinicalTrials.gov Registry (NCT01241318).

## INTRODUCTION

Each year, 140 million newborns are born worldwide.[1] While there has been global progress in reducing child mortality, corresponding rates of reduction in newborn mortality have been far slower and annually 2.5 million newborn deaths occur.[2] As the world strives to meet the Sustainable Development Goals by 2030, it is important to understand not only the direct causes, but also health systems barriers and social determinants of health that lead to poor maternal and newborn outcomes.[3] Factors responsible for these outcomes include poverty, illiteracy, issues of women's empowerment and systematic marginalisation of communities on the

basis of gender, ethnicity or geography.[2 4] In sub-Saharan Africa, birth planning and facility-based childbirth have been core components of interventions to improve maternal and neonatal outcomes. However, this has been insufficient as health systems need to be structurally redesigned to respond to local contexts.[5] Given the emphasis on birth planning in developing countries, it is important to understand why women deliver in locations that differ from what they intended, and what contributes to that change in plan, in order to adequately design interventions and messaging to address specific barriers that prevent women from delivering where they intended.[6–8]

Health facility delivery at primary health facilities with support from referral-level facilities for complicated births has been shown to improve maternal and neonatal outcomes.[5 7 9–11] In order to better understand some of the underlying factors and determinants of access to health facilities equipped with emergency obstetrical and newborn care (EmONC) and skilled birth attendants in rural Zambia, we used data from the Zambia Chlorhexidine Application Trial (ZamCAT), a cluster randomised controlled trial that assessed whether daily 4% chlorhexidine cord cleansing was more effective than dry cord care for prevention of neonatal deaths and omphalitis (umbilical cord infection) in Southern Province.[12] The aim of this prespecified, secondary analysis was to determine the proportion of women who did not deliver where they had intended, to understand the underlying reasons for the discordance between planned and actual delivery locations; and to determine sociodemographic characteristics associated with where pregnant women intended to deliver (ie, concordance with intention and practice).

## METHODS

### Study site

Southern Province, 1 of 10 provinces in Zambia, consists of 13 districts with an estimated population of 2.1 million in 2020.[13] At the time of the ZamCAT Study, the 2013–2014 Zambia Demographic Health Survey (ZDHS) reported that nearly all women (96%) received at least one antenatal care visit during pregnancy.[14] Additionally, more than half of women (55.9%) in Southern Province delivered in a health facility (predominately public sector facilities), while 41% delivered at home.[14] Skilled birth attendants were present for 55% of all deliveries in Southern Province, while untrained family or relatives assisted with 22% of home deliveries.[14]

### Study design

ZamCAT, conducted from 2011 to 2013, was a cluster randomised trial that enrolled and followed a cohort of 39 679 pregnant women living in six districts of Southern Province from before delivery to 28 days after delivery. The analysis presented here was an a priori research question focusing on one of the ZamCAT Study's secondary outcomes, namely: where did pregnant women enrolled in ZamCAT plan to deliver, what were their reasons for

their intended location and what was their actual practice? Our analysis also assessed factors that may have influenced intended and actual delivery locations.

In the main trial, the units of randomisation (clusters) for ZamCAT were Zambian government-run or mission-run primary healthcare centres in Southern Province. Ninety clusters were randomised into two groups: (1) clean dry cord care in accordance with Ministry of Health guidelines (control) and (2) daily 4% chlorhexidine cord washes until 3 days after the umbilical cord fell off (intervention). The study location included a wide range of health facilities, both urban and rural, with differences in staffing and catchment area population characteristics.

Eligible women were in their second or third trimester, aged 15 years or older, planning to stay in the catchment area until 28 days post partum, and willing to provide informed consent and complete cord care as per cluster assignment.[12] Women were screened by field monitors (data collectors), and women were enrolled either at a facility-based antenatal care visit or through community outreach events. At enrolment, a detailed questionnaire was administered which included questions on household information, birth plan and antenatal care characteristics. After the enrolment survey was completed, participants were encouraged to deliver at the nearest health facility and were given standard newborn care messages which included information about delivery location, breast feeding, cord care and danger signs of ill health in their newborn baby as per national and Ministry of Health guidelines. At a home visit 2 weeks after enrolment, all participants, irrespective of study allocation, were provided a standard clean delivery kit. A birth notification system was developed in collaboration with the community specific to each cluster to alert study team of births.[15] For both study arms, field monitors conducted routine home visits at least once during the antenatal period and four times after delivery (days 1, 4, 10 and 28 post partum). Detailed information on where the woman delivered and why was collected during the day 4 postpartum visit. Data were collected using paper forms designed in the Teleforms system (HP Cambridge, UK). Supervisors reviewed forms for illegible and missing entries and worked with the field monitors to make sure they were complete. The forms were transported to the central study office in Choma where a data management team scanned and entered them into a Microsoft Access Database.

### Statistical analysis

Using the full cohort of women enrolled in the study, we assessed women's socioeconomic and pregnancy-related characteristics, and then related them to the outcome of interest (delivery where intended). Our primary outcome was a four-level categorical variable: intended to deliver at health facility and did, intended to deliver at home and did, intended to deliver at a health facility but delivered at home, and intended to deliver at home but delivered

at a health facility. Socioeconomic and pregnancy-related characteristics were documented during the enrolment visit and compared across the intention/practice groups. Pregnancy-related characteristics potentially associated with the outcomes were compared between the two groups using the χ2 test, Fisher's exact test, t-tests or non-parametric Wilcoxon rank-sum tests, as appropriate for continuous or categorical variables. We also assessed the reasons women indicated for why they intended to deliver at a certain location and reasons given after actual delivery on where their actual birth location; women could choose among eight different reasons for why they chose their intended or actual delivery location. Women were allowed to choose more than one reason for why they chose their intended or actual delivery location. All statistical analyses were completed using SAS University Edition.[16] We assessed the association of sociodemographic characteristics with women delivering where they intended to deliver or elsewhere using logistic regression modelling. We also conducted a stratified logistic regression model to assess the predictors of women delivering where they intended stratified by actual delivery location (home and facility).

## Patient and public involvement

Study preparations for the overall ZamCAT required baseline formative ethnographic research, substantial community sensitisation and engagement with three levels of stakeholders, each necessitating different strategies. The process of community engagement for ZamCAT is detailed further by Hamer *et al*.[15] Community leaders and neighbourhood health committees in the 90 clusters identified appropriate field monitor (data collector) candidates according to specific selection criteria. The local health facility staff assisted by ranking lists of potential data collectors provided by the community. Further, the effective birth notification system was developed by the community leaders. Results from the primary study were shared through community meetings in collaboration with the district health offices and at the Zambian National Health Research Conference.

## RESULTS

A total of 39 679 pregnant women were enrolled in ZamCAT between 15 February 2011 and 30 January 2013.[12] During the study, 38 131 women had 38 579 deliveries; 1548 enrolled women were excluded as a result of false pregnancies, miscarriages or abortion, withdrawal from study before delivery, lost to follow-up or death. We had intended and actual delivery information on 37 105 of the enrolled population. The mean age (±SD) of the women was 25.6±6.9 years, 10% had no formal education and 52% had only a primary education (table 1). Of the 37 105 with birth plans stated, 467 (1.2%) had no plan, 2278 (6.1%) planned to deliver at home, 33 953 (91.5%) intended to deliver at a health facility and 407 (1.1%) stated other. In practice, 13 139 (35.4%) actually delivered

at home, 23 666 (63.8%) actually delivered at a health facility and 300 (0.8%) delivered in another location.

## Reasons for planned versus actual delivery location

At study enrolment, respondents were asked about their birth plan and where they planned to deliver as well as why (table 2). Almost all women had a delivery location in their plan (98.8%) and nearly all women (91.6%) reported that they planned to deliver in a health facility. Women who planned to deliver in a health facility stated they were motivated by safety for mother/baby (83.9%) and need for skilled attendance (81.3%). Among those who intended to deliver at home, distance was the most commonly noted reason (37.9%) and same location as previous deliveries (29.2%).

After delivery, women were asked about where they actually delivered and why (table 3). The primary motivations for delivery in the facility were need for skilled attendance (54.6%) and safety for the mother and baby (55.1%). While only 6% of women intended to deliver at home, over one-third (35.4%) of women actually did (N=13 139). The top three reasons for delivering at home were distance (29.7%), need for skilled attendance (43.0%) and safety for mother/baby (44.7%).

## Intended versus actual delivery location

Overall, 67% of women delivered in their intended location. Almost two-thirds of women (n=22 651, 61.1%) who intended to deliver at a health facility actually did (concordant); in contrast, almost one-third of women who intended to deliver at a health facility (n=11 016, 29.7%) delivered at home (discordant). Of those who intended to deliver at home, only 4.1% (n=1515) actually did. Additionally, of those women who intended to deliver at home, 2.0% (n=755) delivered instead at a health facility. Excluded were those who indicated 'no plan' or 'other' for intended delivery location and those who indicated 'other' for actual delivery location.

We observed that 18.7% of those who intended to deliver at a heath facility and did were primiparous, compared with women who intended to deliver at a health facility but delivered at home (19.9%). For those women who were multiparous, more than half (52.7%) intended to deliver at home and actually did; two-thirds (64.3%) of women who intended to deliver at a health facility delivered at home. We found that time between water breaking and delivering in less than 24 hours was not shorter for those who intended to deliver at a health facility but delivered at home. Women who intended to deliver at home and subsequently delivered at a health centre stated they needed a skilled attendant for the safety of the mother and baby (table 4). About 30% of respondents, who indicated distance as a reason for delivering where they did, delivered at home. For those who intended to deliver at a health facility and did, 41% lived less than a 1-hour walk from the health facility. Of women who intended to deliver at home and did, 64% lived between 1 and 3 hours' walk from the nearest health

**Table 1**  Demographic, household and pregnancy characteristics of women enrolled in ZamCAT

| Characteristics | Intended to deliver at health facility and did<br>n=22 651 | Intended to deliver at home and did<br>n=1515 | Intended to deliver at health facility but delivered at home<br>n=11 016 | Intended to deliver at home but delivered at a health facility<br>n=755 |
|---|---|---|---|---|
| **Maternal age (years), n (%)** | | | | |
| <20 | 5919 (26.1) | 185 (12.2) | 2115 (19.2) | 215 (28.5) |
| 20–35 | 14 473 (63.9) | 1114 (73.5) | 7626 (69.2) | 442 (58.5) |
| >35 | 2259 (10.0) | 216 (14.3) | 1275 (11.6) | 98 (13.0) |
| Missing | | | | |
| **Tribe, n (%)** | | | | |
| Tonga | 19 295 (85.2) | 1431 (94.5) | 10 257 (93.1) | 670 (88.7) |
| Ila | 142 (0.6) | 7 (0.5) | 53 (0.5) | 4 (0.5) |
| Lozi | 1175 (5.2) | 20 (1.3) | 238 (2.2) | 28 (3.7) |
| Nyanja | 687 (3.0) | 18 (1.2) | 147 (1.3) | 13 (1.7) |
| Bemba | 568 (2.5) | 23 (1.5) | 135 (1.2) | 24 (3.2) |
| Other | 781 (3.4) | 15 (1.0) | 183 (1.7) | 12 (1.6) |
| Missing | 3 (0.08) | 1 (0.1) | 3 (0.03) | 4 (0.5) |
| **Women's education level, n (%)** | | | | |
| No education | 1681 (7.4) | 265 (17.5) | 1442 (13.1) | 95 (12.6) |
| Some primary education | 10 779 (47.6) | 1001 (66.1) | 6487 (58.9) | 366 (48.5) |
| Some secondary education | 9884 (43.6) | 246 (16.2) | 3067 (27.8) | 290 (38.4) |
| Higher than secondary education | 288 (1.3) | 2 (0.13) | 14 (0.1) | 0 (0.0) |
| Unknown or missing | 19 (0.1) | 1 (0.1) | 6 (0.06) | 4 (0.5) |
| **Maternal literacy, n (%)** | | | | |
| Illiterate | 4875 (21.5) | 611 (40.3) | 3720 (33.8) | 226 (29.9) |
| Can read a bit | 10 620 (46.9) | 716 (47.3) | 5212 (47.4) | 332 (44.0) |
| Can read very well | 7078 (31.2) | 177 (11.7) | 2022 (18.4) | 191 (25.3) |
| No response or missing | 78 (0.3) | 11 (0.7) | 62 (0.6) | 6 (0.8) |
| **Wealth index, n (%)** | | | | |
| 1—lowest | 5214 (23.0) | 541 (35.7) | 3154 (28.6) | 246 (32.6) |
| 2 | 5407 (23.9) | 421 (27.8) | 2910 (26.4) | 175 (23.2) |
| 3 | 5490 (24.2) | 361 (23.8) | 2807 (25.5) | 181 (24.0) |
| 4—highest | 6540 (28.9) | 192 (12.7) | 2145 (19.5) | 151 (20.0) |
| **Marital status, n (%)** | | | | |
| Single | 4173 (18.4) | 82 (5.4) | 1243 (11.3) | 132 (17.5) |
| Married or cohabitating | 18 267 (80.7) | 1405 (92.7) | 9644 (87.6) | 600 (79.5) |
| Separated, divorced or widowed | 204 (0.9) | 26 (1.7) | 128 (1.2) | 19 (2.5) |
| Missing | 7 (0.03) | 2 (0.1) | 1 (0.01) | 4 (0.5) |
| **Median household size (IQR)** | 6 (4–8) | 6 (4–8) | 6 (4–8) | 6 (4–8) |
| **Time walking from home to health facility** | | | | |
| <1 hour | 9276 (41.0%) | 293 (19.3%) | 2301 (20.9%) | 253 (33.5%) |
| 1–<2 hours | 7823 (34.5%) | 526 (34.7%) | 4202 (38.1%) | 278 (36.8%) |
| 2–<3 hours | 3981 (17.6%) | 449 (29.6%) | 3054 (27.7%) | 140 (18.5%) |
| 3–<4 hours | 1047 (4.6%) | 167 (11.0%) | 1017 (9.2%) | 58 (7.7%) |
| 4–<5 hours | 250 (1.1%) | 43 (2.8%) | 248 (2.3%) | 12 (1.6%) |
| 5 hours or more | 120 (0.5%) | 12 (0.8%) | 129 (1.2%) | 8 (1.1%) |

Continued

**Table 1** Continued

| Characteristics | Intended to deliver at health facility and did<br>n=22 651 | Intended to deliver at home and did<br>n=1515 | Intended to deliver at health facility but delivered at home<br>n=11 016 | Intended to deliver at home but delivered at a health facility<br>n=755 |
|---|---|---|---|---|
| Unknown | 81 (0.4%) | 17 (1.1%) | 42 (0.4%) | 2 (0.3%) |
| Missing | 74 (0.3%) | 8 (0.5%) | 23 (0.2%) | 4 (0.5%) |
| **Pregnancy characteristics** | | | | |
| Parity, mean (SD) | 2.7 (2.3) | 2.4 (2.3) | 2.3 (2.3) | 2.5 (2.3) |
| Primiparous, n (%) | 4230 (18.7) | 231 (15.3) | 2188 (19.9) | 119 (15.8) |
| Multiparous, n (%) | 11 939 (52.7) | 1131 (74.7) | 7086 (64.3) | 401 (53.1) |
| Missing, n (%) | 6482 (28.6) | 153 (10.1) | 1742 (15.8) | 235 (31.1) |
| Gravida, mean (SD) | 3.8 (2.4) | 3.5 (2.3) | 3.4 (2.3) | 3.5 (2.3) |
| Primigravida, n (%) | 6327 (27.9) | 148 (9.8) | 1702 (15.5) | 225 (29.8) |
| Multigravida, n (%) | 16 294 (71.4) | 1362 (89.9) | 9298 (84.4) | 525 (69.5) |
| Missing, n (%) | 30 (0.1) | 5 (0.3) | 16 (0.2) | 5 (0.7) |
| Gestational age at enrolment, mean (SD) | 27.9 (6.8) | 28.0 (7.0) | 28.1 (7.1) | 28.1 (7.0) |
| Time between water breaking and delivery, n (%) | | | | |
| Less than 24 hours | 20 259 (91.7) | 1344 (88.7) | 10 123 (91.9) | 418 (55.4) |
| 24–48 hours | 781 (3.5) | 25 (1.7) | 168 (1.5) | 35 (4.6) |
| More than 48 hours | 120 (0.54) | 8 (0.53) | 26 (0.2) | 2 (0.3) |
| Don't know | 915 (4.1) | 120 (8.0) | 446 (4.1) | 32 (4.2) |
| Missing | 576 (2.5) | 1 (1.2) | 253 (2.3) | 268 (35.5) |

Those who responded to having 'no plan' or 'other' when asked about intended delivery location, and those who responded as 'other' when asked about actual delivery location were excluded from demographic analysis (n=1168).
ZamCAT, Zambia Chlorhexidine Application Trial.

facility. Among those who intended to deliver at a health facility and actually delivered at home, 46.3% indicated that they did so because of safety for mother/baby. Additionally, this same group indicated a need for skilled attendance as a reason for delivering at home versus at a health facility. Among those who intended to and actually delivered at home, only 8.9% indicated family/social

**Table 2** Planned delivery location and rationale

| | No plan (N=467) | | Health facility (N=33 953) | | Home (N=2278) | | Other (N=407) | |
|---|---|---|---|---|---|---|---|---|
| | n | % | n | % | n | % | n | % |
| Same location as prior delivery | 3 | 0.6 | 4237 | 12.5 | 666 | 29.2 | 55 | 13.5 |
| Need for skilled attendance | 14 | 3.0 | 27 578 | 81.2 | 402 | 17.7 | 281 | 69.0 |
| Financial restraints | 13 | 2.8 | 1058 | 3.1 | 443 | 19.5 | 16 | 3.9 |
| Distance | 10 | 2.1 | 4714 | 13.9 | 851 | 37.4 | 45 | 11.1 |
| Relationship with providers | 3 | 0.8 | 453 | 1.3 | 175 | 7.7 | 8 | 2.0 |
| Family/social expectations | 2 | 0.5 | 714 | 2.1 | 264 | 11.6 | 2 | 0.5 |
| Safety for mother/baby | 12 | 2.6 | 28 548 | 84.1 | 464 | 20.4 | 293 | 72.0 |
| Other | 3 | 0.6 | 276 | 0.8 | 47 | 2.1 | 33 | 8.1 |
| Missing | 0 | 0.0 | 1 | 0.0 | 5 | 0.2 | 4 | 1.0 |

'Other' in birth plan often includes hospital delivery which is separate from health facility delivery as well as TBA delivery.
TBA, traditional birth attendant.

**Table 3** Actual delivery location and reasoning

| | Health facility (N=23 666) | | Home (N=13 139) | | Other (N=300) | |
|---|---|---|---|---|---|---|
| | n | % | n | % | n | % |
| Same location as prior delivery | 3182 | 13.5 | 2062 | 15.7 | 47 | 15.7 |
| Need for skilled attendance | 12 916 | 54.6 | 5654 | 43.0 | 164 | 54.7 |
| Financial restraints | 1799 | 7.6 | 2234 | 9.4 | 13 | 4.3 |
| Distance | 6303 | 26.6 | 3898 | 29.7 | 74 | 24.7 |
| Relationship with providers | 571 | 2.4 | 426 | 3.2 | 9 | 3.0 |
| Family/social expectations | 1251 | 5.3 | 961 | 7.5 | 17 | 5.7 |
| Safety for mother/baby | 13 048 | 55.1 | 5866 | 44.7 | 170 | 56.7 |
| Other | 1468 | 6.2 | 792 | 6.0 | 23 | 7.7 |
| Missing | 1353 | 5.7 | 762 | 5.8 | 15 | 5.0 |

Health facility in actual delivery location includes hospital delivery.

expectations as a reason while 34.5% indicated safety for mother/baby as a reason.

### Predictors of women delivering where intended

Table 5 shows the result of our stratified logistic regression model where we assessed predictors of women delivering where they intended to, stratified by home delivery and facility delivery. Among women who delivered at home, we found that women who were aged 25–34 and ≥35 years were more likely to deliver where they intended than women aged 20–24 years (adjusted OR (aOR)=1.31 (95% CI=1.14 to 1.50), p=0.0001 and aOR=1.32 (95%

CI=1.11 to 1.57), p=0.001, respectively). Women who were 15–19 years and delivered at home had lower odds (aOR=0.77 (95% CI=0.60 to 0.93), p=0.007) of delivering where they intended compared with those aged 20–24 years. Women who delivered at home and had more than a primary education were less likely to deliver where they intended when compared with those with no education (aOR=0.55 (95% CI=0.43 to 0.69), p<0.0001). Those who delivered at home and were not in the lowest wealth quintile were less likely to deliver where they intended (table 5). When examining the time to walk to the nearest

**Table 4** Comparison of intended and actual delivery location, with reasons for actual delivery location

| Reasons for actual delivery location | Intended to deliver at health facility and did (N=22 651) | | Intended to deliver at home and did (N=1515) | | Intended to deliver at health facility but delivered at home (N=11 016) | | Intended to deliver at home but delivered at a health facility (N=755) | |
|---|---|---|---|---|---|---|---|---|
| | n | % | n | % | n | % | n | % |
| Same location as prior delivery | 3020 | 13.4 | 279 | 18.4 | 1677 | 15.2 | 120 | 15.9 |
| Need for skilled attendance | 12 458 | 54.5 | 476 | 31.4 | 4947 | 44.9 | 353 | 46.8 |
| Financial restraints | 1708 | 7.7 | 149 | 9.8 | 1021 | 9.3 | 72 | 9.5 |
| Distance | 5987 | 26.6 | 505 | 33.3 | 3229 | 29.3 | 233 | 30.9 |
| Relationship with providers | 528 | 2.4 | 76 | 5.0 | 326 | 3.0 | 34 | 4.5 |
| Family/social expectations | 1182 | 5.3 | 135 | 8.9 | 782 | 7.1 | 48 | 6.4 |
| Safety for mother/baby | 12 585 | 36.9 | 523 | 34.5 | 5095 | 46.3 | 332 | 44.0 |
| Other | 1408 | 6.2 | 107 | 7.1 | 654 | 5.9 | 44 | 5.8 |
| Missing | 1296 | 6.2 | 89 | 5.9 | 628 | 5.7 | 47 | 6.2 |
| **Time walking from home to health facility** | | | | | | | | |
| <1 hour | 9276 | 41.0 | 293 | 19.3 | 2301 | 20.9 | 253 | 33.5 |
| 1–<2 hours | 7823 | 34.5 | 526 | 34.7 | 4202 | 38.1 | 278 | 36.8 |
| 2–<3 hours | 3981 | 17.6 | 449 | 29.6 | 3054 | 27.7 | 140 | 18.5 |
| 3–<4 hours | 1047 | 4.6 | 167 | 11.0 | 1017 | 9.2 | 58 | 7.7 |
| 4–<5 hours | 250 | 1.1 | 43 | 2.8 | 248 | 2.3 | 12 | 1.6 |
| 5 hours or more | 120 | 0.5 | 12 | 0.8 | 129 | 1.2 | 8 | 1.1 |
| Unknown | 81 | 0.4 | 17 | 1.1 | 42 | 0.4 | 2 | 0.3 |
| Missing | 74 | 0.3 | 8 | 0.5 | 23 | 0.2 | 4 | 0.5 |

**Table 5** Predictors of women delivering where they intended to, stratified by delivery location

| Characteristics | Among those who delivered at home, adjusted OR (aOR) of delivering where intended | | Among those who delivered at facility, aOR of delivering where intended | |
|---|---|---|---|---|
| | aOR (95% CI) | P value | aOR (95% CI) | P value |
| **Maternal age (years)** | | | | |
| 15–19 | 0.77 (0.64 to 0.93) | **0.007** | 0.69 (0.57 to 0.82) | **<0.0001** |
| 20–24 | Reference | | Reference | |
| 25–34 | 1.31 (1.14 to 1.50) | **0.0001** | 0.89 (0.75 to 1.05) | **0.17** |
| >35 | 1.32 (1.11 to 1.57) | **0.001** | 0.73 (0.59 to 0.90) | **0.003** |
| **Women's education level** | | | | |
| No education | Reference | | Reference | |
| Any primary | 0.90 (0.75 to 1.07) | 0.21 | 1.34 (1.09 to 1.72) | **0.008** |
| More than primary | 0.55 (0.43 to 0.69) | **<0.0001** | 1.54 (1.17 to 2.02) | **0.002** |
| **Marital status** | | | | |
| Married | Reference | | Reference | |
| Not married | 0.67 (0.55 to 0.83) | **0.0002** | 0.89 (0.76 to 1.06) | 0.18 |
| **Can read** | | | | |
| No | Reference | | Reference | |
| Yes | 1.05 (0.91 to 1.20) | 0.50 | 1.33 (1.11 to 1.58) | **0.002** |
| **Household light source** | | | | |
| No electricity | Reference | | Reference | |
| Electricity | 0.81 (0.57 to 1.15) | 0.24 | 1.27 (0.95 to 1.69) | 0.11 |
| **Wealth index** | | | | |
| 1—lowest | Reference | Reference | Reference | Reference |
| 2 | 0.85 (0.74 to 0.98) | **0.02** | 1.34 (1.13 to 1.60) | **0.0008** |
| 3 | 0.76 (0.66 to 0.88) | **0.0002** | 1.31 (1.10 to 1.55) | **0.002** |
| 4—highest | 0.56 (0.46 to 0.67) | **<0.0001** | 1.65 (1.36 to 2.01) | **<0.0001** |
| **Primigravida†** | | | | |
| No | Reference | | Reference | |
| Yes | 0.90 (0.73 to 1.11) | 0.34 | 0.63 (0.52 to 0.76) | **<0.0001** |
| **Time to walk to nearest health facility†** | | | | |
| <1 hour | 1.10 (0.94 to 1.28) | 0.22 | 1.11 (0.95 to 1.30) | 0.20 |
| 1–<2 hours | Reference | | Reference | Reference |
| 2–<3 hours | 1.17 (1.02 to 1.33) | **0.03** | 0.96 (0.80 to 1.14) | 0.61 |
| 3–<4 hours | 1.27 (1.05 to 1.54) | **0.01** | 0.73 (0.56 to 0.95) | **0.02** |
| 4–<5 hours | 1.20 (0.84 to 1.68) | 0.32 | 0.77 (0.46 to 1.28) | 0.31 |
| 5 hours or more | 0.68 (0.37 to 1.23) | 0.20 | 0.67 (0.32 to 1.38) | 0.27 |
| **Time between water breaking and delivery†** | | | | |
| Less than 24 hours | Reference | | Reference | |
| 24–48 hours | 1.04 (0.68 to 1.59) | 0.87 | 0.54 (0.39 to 0.73) | **<0.0001** |
| More than 48 hours | 2.11 (0.95 to 4.69) | 0.07 | 1.29 (0.41 to 4.09) | 0.66 |
| Don't know | 1.93 (1.56 to 2.38) | **<0.0001** | 0.47 (0.35 to 0.61) | **<0.0001** |
| **Complications during pregnancy*†** | | | | |
| High fever | 1.45 (1.00 to 2.11) | 0.05 | 0.76 (0.52 to 1.11) | 0.15 |
| Bleeding | 1.23 (0.80 to 2.05) | 0.31 | 0.60 (0.40 to 0.90) | **0.01** |

Continued

**Table 5** Continued

| Characteristics | Among those who delivered at home, adjusted OR (aOR) of delivering where intended | | Among those who delivered at facility, aOR of delivering where intended | |
| --- | --- | --- | --- | --- |
| | aOR (95% CI) | P value | aOR (95% CI) | P value |
| Fits | 2.15 (0.79 to 5.82) | 0.13 | 0.39 (0.19 to 0.81) | **0.02** |
| Headache | 1.07 (0.79 to 1.44) | 0.67 | 1.71 (1.11 to 2.62) | **0.01** |
| Vaginal discharge | 0.50 (0.15 to 1.61) | 0.24 | 0.23 (0.13 to 0.40) | **<0.0001** |
| Heavy bleeding | 1.08 (0.78 to 1.50) | 0.63 | 0.71 (0.54 to 0.94) | **0.02** |
| Swelling | 0.88 (0.63 to 1.22) | 0.43 | 0.82 (0.61 to 1.10) | 0.20 |

Significant *p*-values in **bold**.
*Referent groups for complications were the absence of specified complications.
†Adjusted for maternal age, women's education, marital status, literacy, wealth index, housing light source.

health facility, we found that those who delivered at home had higher odds of delivering where they intended to if they lived 2 to less than 3 hours from the nearest health facility (aOR=1.17 (95% CI=1.02 to 1.33), p=0.03) or 3 to less than 4 hours away from the nearest health facility (aOR=1.27 (95% CI=1.05 to 1.54), p=0.01).

Women who delivered at the facility had greater odds of delivering where they intended if they received any primary schooling (aOR=1.34 (95% CI=1.09 to 1.72), p=0.008) or more than a primary school education (aOR=1.54 (95% CI=1.17 to 2.02), p=0.002), were literate (aOR=1.33 (95% CI=1.11 to 1.58), p=0.002), were in the second (aOR=1.34 (95% CI=1.13 to 1.60), p=0.0008), third (aOR=1.31 (95% CI=1.10 to 1.55), p=0.002), or fourth (aOR=1.65 (95% CI=1.36 to 2.01), p<0.0001) quintile of the wealth index, and those who experienced headache as a complication during pregnancy (aOR=1.71 (95% CI=1.11 to 2.62), p=0.01). In contrast, women who delivered at a health facility had lower odds of delivering where they intended if it was their first pregnancy (aOR=0.63 (95% CI=0.52 to 0.76), p<0.0001) compared with those who had been pregnant before, lived between a 3 to less than 4-hour walk from the nearest health facility (aOR=0.73 (95% CI=0.56 to 0.95), p=0.02) compared with those who lived a 1 to less than 2-hour walk, lapsed 24–48 hours between water breaking and delivery (aOR=0.54 (95% CI=0.39 to 0.73), p<0.0001) compared with those who only lapsed less than 24 hours, and those who experienced bleeding (aOR=0.60 (95% CI=0.40 to 0.91), p=0.01), fits (aOR=0.39 (95% CI=0.19 to 0.81), p=0.01), vaginal discharge (aOR=0.23 (95% CI=0.13 to 0.40), p<0.0001), and/or heavy bleeding (aOR=0.71 (95% CI=0.54 to 0.94), p=0.02) compared with those who did not experience these symptoms.

## DISCUSSION

Overall, birth plans were widely prepared in this cohort and >90% of women indicated a plan to deliver at a health facility, consistent with the 2013–2014 ZDHS.[14] However, only two-thirds of women had actual delivery location in line with their birth plan and >35% of women delivered at home. Our analysis demonstrates clear deviations from the birth plan during the actual delivery, especially among those who had expressed an intent to deliver at a facility. Discrepancy between planning and actual delivery behaviour highlights that the development of birth plans and exposure to birth planning messaging are insufficient to address barriers to adhering to plans to have a facility-based childbirth.

Several reasons may underlie the discrepancy between intended location and actual practice around childbirth. One potential reason could be due to women's increased concerns about safety at facilities that are poorer quality. Interestingly, 46% of women in our sample indicated safety for mother/baby as a reason for why they delivered at home even though they intended to deliver at a health facility. Unfortunately, we do not know if this was driven by fear of poor treatment at the facility or fear of transporting/moving the labouring women. In some instances, poorly functioning health systems are unable to engage women in antenatal and postnatal care to create sufficient continuity of care.[17] Some of these barriers may relate to a sense of fatalism around maternal and newborn care, and are important contributors to preventable mortality.[18] Such disparities present a major challenge to policies and programmes aiming to mitigate the burden of maternal and neonatal mortality. The large proportion of facility births in unclean, unhygienic environments, and poor postnatal care may be important contributors to why women indicate safety for mother/baby as a reason why they changed their birth plan from intending to deliver at a health facility to delivering at home.

Another factor that may contribute to discrepancy between intention and practice is the dynamic between provider and patient.[19–21] Among those who intended to deliver at a health facility but instead delivered at home, 45% indicated need for skilled delivery as a reason. While this seems counterintuitive, one reason for this may be the presence of traditional birth attendants (TBAs) in the community who women regard as skilled attendants.[22 23] Although a policy change in Zambia resulted in TBAs no longer being trained and recommended that they stop conducting deliveries at home, a study on reasons for home delivery and use of TBAs in rural Zambia by Sialubanje *et al* found that most women had positive attitudes

towards TBAs and regarded them as respectful, skilled and trustworthy.[19] Moreover, most women indicated that negative attitudes toward the quality of services at the clinic as a reason for why they delivered at home.[19] Smith *et al* focused on behavioural drivers in labour and delivery wards in Chipata District in Eastern Province, Zambia and found that women in the wards are often subjected to disrespect and abuse.[24] Another study in rural Kenya found that although the community was aware of the risks of delivering with a TBA, many still preferred to deliver with a TBA due to their availability, accessibility and their friendly attitude.[22] In Zambia, a study by Biemba *et al,* which used data from the ZamCAT Study, found that TBAs often end up tending to deliveries even in health facilities due to a lack of qualified personnel.[25] As for the safety of mother and child, some women perceive delivering at the health facility as unsafe based on cultural values of elder women in the community.[26] Finally, in another study, women in Ethiopia who had low perceived susceptibility and severity were more likely to intend to deliver at home instead of at a health facility.[26 27]

Among women who intended to deliver at home, distance was most often cited as a reason (37.9%), although this less often reported when women were asked why they delivered home post-delivery. Our data are similar to the 2013–2014 ZDHS which reported that 32.2% of women in Southern Province indicated distance and lack of transportation as a reason for not delivering in a health facility. Additionally, a study in Ebonyi State, Nigeria on the choice of birth place among antenatal clinic attendees in rural mission hospitals also found that distance served as a major determinant of birth place location.[28] Distance as a barrier to accessing care is not uncommon in many resource-constrained countries in sub-Saharan Africa and Asia.[29 30] Scott *et al* found that women who lived in remote areas of Zambia faced persistent challenges regarding delivery location.[31] The study which included four districts in Southern Province also found women who lived far from a health facility, attended fewer antenatal visits, were multigravida and who did not use maternity waiting homes had higher odds of home delivery.[31] Maternal waiting homes have shown to increase likelihood of delivering at a health facility.[31–33] Given that most maternal waiting homes were renovated or built after 2013, our study did not collect data on how maternal waiting home influenced delivery locations for those further away from the health facility.

Our analysis is strengthened by the sample size and variation in adherence to intended delivery location. However, a key limitation is a likelihood for social desirability bias among respondents as questions such as birth plan and reasons behind delivery choice may have influenced women to respond why they intended to deliver in a facility. Despite the multiple messages provided to the participants during the antenatal period encouraging them to deliver in health facilities, a substantial number of women were unable or opted not to deliver at a facility. The decision on where to deliver may have been made in consultation with family members or influenced by other forces which were missed in our data capture. Another limitation is that the data are 7 years old. Although our findings reflected facility-based childbirth trends around 2014,[14] facility-based childbirth in Zambia by 2018 had increased.[34] However, given the recent COVID-19 pandemic, facility-based childbirth has again decreased and our findings may be applicable.[34 35] Finally, we do not have qualitative data that could have helped explore the underlying motivations around intention and actual delivery locations.

Understanding factors that influence women to change their birth plan as well as addressing structural and perceived barriers to facility delivery are important to improving care. Maternal and neonatal outcomes can be significantly improved if women delivered at a facility with at least basic EmONC.[36 37] Potential interventions that can address barriers to reaching facilities include provision of maternity waiting homes, addressing perceived safety and treatment of mothers and infants in facilities, supporting providers to provide high-quality/respectful maternity care, system-wide plans that support transport of pregnant women to a health facility, messaging around saving for transport to a health facility, further health education for those who are primigravida to deliver a health facility and safe delivery incentive programmes.

**Author affiliations**
[1]Doctor of Public Health Program, Boston University School of Public Health, Boston, Massachusetts, USA
[2]Department of Global Health and Population, Harvard University T H Chan School of Public Health, Boston, Massachusetts, USA
[3]Department of Global Health, Boston University School of Public Health, Boston, Massachusetts, USA
[4]Department of Pediatrics, Boston University School of Medicine, Boston, Massachusetts, USA
[5]Public Health Unit, Fr Thomas Alan Rooney Memorial Hospital, Kumasi, Asankrangwa, Ghana
[6]Zambia National Health Research Authority, Lusaka, Zambia
[7]Levy Mwanawasa Medical University, Lusaka, Zambia
[8]MBBS Department, Dow University of Health Sciences, Karachi, Pakistan
[9]Section of Infectious Disease, Department of Medicine, Boston University School of Medicine, Boston, Massachusetts, USA
[10]Ariadne Labs, Brigham and Women's Hospital, Boston, Massachusetts, USA
[11]Division of Global Health Equity & Department of Medicine, Brigham and Women's Hospital, Boston, Massachusetts, USA
[12]Department of Medicine, Harvard Medical School, Boston, MA, USA

**Contributors** As the original co-investigators of the ZamCAT, DH and KEAS were responsible for conceiving the original research question, designing the trial and overseeing the study implementation. KEAS, DH and HS were responsible for the conceptualisation of this subanalysis. HS was responsible for the methodology, conducting formal analysis, and the writing and revision of the manuscript. JH, GB, KM and KY-A were part of the original research team for ZamCAT and were responsible for reviewing and editing the manuscript. EH was responsible for dataset creation, preliminary analysis and manuscript review. AB assisted with preliminary analysis, reviewing and editing of the manuscript. HS is responsible for the overall content as guarantor.

**Funding** This work was supported, in whole or in part, by the Bill & Melinda Gates Foundation OPPGH5298. Under the grant conditions of the Foundation, a Creative Commons Attribution 4.0 Generic License has already been assigned to the Author Accepted Manuscript version that might arise from this submission. This secondary

analysis research received no specific grant from any funding agency in the public, commercial or not-for-profit sectors. The original data collection and primary trial was funded by the Bill & Melinda Gates Foundation (global health grant number OPPGH5298).

**Disclaimer** The findings and conclusions contained within are those of the authors and do not necessarily reflect positions or policies of the Bill & Melinda Gates Foundation.

**Map disclaimer** The inclusion of any map (including the depiction of any boundaries therein), or of any geographic or locational reference, does not imply the expression of any opinion whatsoever on the part of BMJ concerning the legal status of any country, territory, jurisdiction or area or of its authorities. Any such expression remains solely that of the relevant source and is not endorsed by BMJ. Maps are provided without any warranty of any kind, either express or implied.

**Competing interests** None declared.

**Patient and public involvement** Patients and/or the public were involved in the design, or conduct, or reporting, or dissemination plans of this research. Refer to the Methods section for further details.

**Patient consent for publication** Not required.

**Ethics approval** The Boston University Medical Campus Institutional Review Board (FWA# 00008404) and University of Zambia Research Ethics Committee (FWA# 00001131) provided ethical clearance. The Zambia National Health Research Authority approved the study. All women provided written informed consent, which was obtained in English or Tonga.

**Provenance and peer review** Not commissioned; externally peer reviewed.

**Data availability statement** Data are available upon reasonable request. The data that support the findings of this study are available from the corresponding author, upon reasonable request.

**ORCID iDs**
Hiwote Solomon http://orcid.org/0000-0001-9125-5217
Kojo Yeboah-Antwi http://orcid.org/0000-0002-3516-9266
Davidson H Hamer http://orcid.org/0000-0002-4700-1495
Katherine E A Semrau http://orcid.org/0000-0002-8360-1391

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
