## [Reviewer comments · BMJ Open]

ARTICLE DETAILS

TITLE (PROVISIONAL)	Intended versus actual delivery location and factors associated with change in delivery location among pregnant women in Southern Province, Zambia: a pre-specified secondary observational analysis of the ZamCAT trial
AUTHORS	Solomon, Hiwote; Henry, Elizabeth; Herlihy, Julie; Yeboah-Antwi, Kojo; Biemba, Godfrey; Musokotwane, Kebby; Bhutta, Afsah; Hamer, D.; Semrau, Katherine

VERSION 1 – REVIEW

REVIEWER	Adewuyi, Emmanuel Queensland University of Technology
REVIEW RETURNED	19-Aug-2021

GENERAL COMMENTS	I found this manuscript well written, aims and objectives clearly stated, and the rationale justified. The subject of the study is important, interesting and can contribute to the current body of knowledge in maternal child health care. I have a few minor comments. 1. In the abstract: 'Among women who delivered at home ...' (line 35 – 41), the result reported '(aOR=1.31, (95% CI=1.14, 1.50), p=0.17)' the confidence interval suggests the result was significant, but the p-value does not. Why? Same was reported in the results section (page 15, lines 15 – 18). However, the p-value in Table 5 suggests the results was significant. I guess this must have been a mistake which authors need to correct. Given there are many Tables because of the volume of the work done, authors need to double check that there are no similar mistakes in their manuscript. 2. Authors need to re-write the conclusion in the abstract section. There should be a summary of findings and brief recommendations.
--

REVIEWER	Singh, Amarjeet PGIMER, Department of Community Medicine
REVIEW RETURNED	12-Nov-2021

GENERAL COMMENTS	Where to deliver? Intention versus actual practice in pregnant women in Southern Province, Zambia Abstract The aim of this pre-specified (MEANING?), secondary analysis of the Zambia Chlorhexidine Application Trial (ZamCAT) vs Recruitment (of the Trial - clarify) took place at 90 health facilities Ethical aspects??? – Ethically, the investigators should have advised / educated the subjects to deliver at a HF ; please clarify, if this is so? WAS IT DONE? IF YES , IT WOULD AFFEST THE RESULTS! A total of 39,679 pregnant women were enrolled in ZamCAT between Feb 15, 2011 and Jan 30, 2013. – = too old Secondary outcome measures of what ? vs WHAT WAS THE PRIMARY ONE? When interviewed antepartum, 92% of respondents intended to deliver at a HF, 6.1% at home, and 1.2% had no plan. However, of those who intended to deliver at a HF, 61% did; of those who intended to deliver at home, only 4% did; and of those who intended to deliver at home, 2% delivered instead at a HF. Conclusion: DO NOT SEEM TO HAVE EMERGED FROM THE STUDY Understanding factors that influence women to change their birth plan as well as addressing structural and perceived barriers to facility delivery are important to improving care. Article Summary Strengths and limitations of this study:  • In this large, cluster-randomized controlled trial with more than 39,000 pregnant women enrolled, less than 4% of enrolled women were lost before delivery, and less than 1% were lost to follow-up during the neonatal period. DOES NOT SEEM TO HAVE LINKAGE WITH THIS ARTICLE Discrepancies found between intended and actual delivery location highlight that the development of birth plans and exposure to birth planning messaging are insufficient (NOT NECESSARILY) to address barriers to adhering to plans to have a facility-based childbirth. Introduction However, this (WHAT?) has been insufficient as health systems need to be structurally redesigned to respond to local contexts Given the emphasis on birth planning in developing countries (PROOF?; BY WHOM?) Methods
--

The analysis presented here was an a priori research question focusing on the **secondary outcomes ? of the ZamCAT study**: where did pregnant women enrolled in ZamCAT plan to deliver, what were their reasons for their intended location, and what was their actual practice?

Results – THE TOTALS DO NOT MATCH!

Table 1. Demographic, household, and pregnancy characteristics of women enrolled in ZamCAT

Intended to deliver at health facility and did Intended to deliver at home and did Intended to deliver at health facility but delivered at home Intended to deliver at home but delivered at a health facility
Characteristics n=22 651 n=1515 n=11 016 n=755

Table 2. Planned delivery location and rationale

No Plan (N=467) Health Facility (N=33,953) Home (N=2,278)
Other (N=407)

Table 4. Need for skilled attendance ???? TOLD BY 476 (31.4%) / Safety for mother-baby BY 523 (34.5%) AS THE Reason for Actual AND Intended Delivery Location at home ??

Discussion

Interestingly, 46% of women in our **sample indicated safety for mother/baby as a reason for why they delivered at home** even though they intended to deliver at a health facility. Unfortunately, we do not know if this was driven by fear of poor treatment at the facility or fear of transporting/moving the laboring women.

IT COULD HAVE BEEN DUE TO THEIR LACK OF COMPREHENSION ABOUT THE QUESTION!?

Among those who intended to deliver at a health facility but instead delivered at home, 45% indicated need for skilled delivery as a reason. While this seems counterintuitive, one reason for this may be the presence of traditional birth attendants (TBAs) in the community who women regard as skilled attendants

AGAIN, IT COULD HAVE BEEN DUE TO THEIR LACK OF COMPREHENSION ABOUT THE TERM- 'skilled delivery'!?

Although a **policy change (GIVE REFERENCE- WHO/UNFPA/UNICEF)** in Zambia resulted in **TBAs no longer being trained** and recommended that they stop conducting deliveries at home (**IF SO, WHY THE DID TRIAL INCLUDE Chlorhexidine Application at home?**) , a study on reasons for home delivery and use of TBAs in rural Zambia by Sialubanje et al. found that most women had positive attitudes towards TBAs and regarded them as respectful, skilled, and trustworthy.

Moreover, most women indicated that negative attitudes toward the quality of services at the clinic as a reason for why they

	delivered at home. 17 IS THIS SENTENCE FROM REF. NO. 17? OR FROM THIS STUDY...NOT CLEAR?!s Smith et al. focused on behavioral drivers in labor and delivery wards in Chipata District in Eastern Province, Zambia and found that women in the wards are often subjected to disrespect and abuse.²² NO REFERENCE TO LOW STATUS OF WOMEN AS A DETERMINANT in this study WHICH COULD AFFECT the Intention versus actual practice in pregnant women as to Where to deliver? IN NUTSHELL, this SEEMS TO BE A redundant publication! Zambia Chlorhexidine Application Trial (ZamCAT) Primary objectives - NO LINK WITH the Secondary objective listed below  • To determine whether 4% chlorhexidine cord cleansing is more effective than dry cord care for the prevention of neonatal mortality Secondary objectives  • To compare where pregnant women plan to deliver and where they actually deliver
--	---

VERSION 1 – AUTHOR RESPONSE

Reviewer 1:

- In the abstract the following were suggested:
 - ‘Among women who delivered at home ...’ (line 35 – 41), the result reported ‘(aOR=1.31, (95% CI=1.14, 1.50), p=0.17)’ the confidence interval suggests the result was significant, but the p-value does not. Why?
 - **Response:** Thank you for pointing out this inconsistency. This was our error; we have reviewed all tables and text to ensure the numbers are consistent with the analysis.
 - Same was reported in the results section (page 15, lines 15 – 18). However, the p-value in Table 5 suggests the results was significant. I guess this must have been a mistake which authors need to correct.
 - **Response:** Thank you for pointing out this inconsistency. This was our error; we have reviewed all tables and text to ensure the numbers are consistent with the analysis.
 - Given there are many Tables because of the volume of the work done, authors need to double check that there are no similar mistakes in their manuscript.
 - **Response:** Thank you for this suggestion. We have double checked the other figures in all tables including the 95% confidence intervals and p values.
- Authors need to re-write the conclusion in the abstract section. There should be a summary of findings and brief recommendations.
 - **Response:** Thank you for the recommendation, we have updated the abstract.

Reviewer 2:

- Clarification on meaning of “pre-specified secondary analysis”
 - **Response:** The results presented in this manuscript represent an analysis of a secondary objective of the original ZamCAT protocol as outlined in the original protocol and thus constitute an *a priori* or “pre-specified secondary analysis”.
- Ethical considerations of trial, specifically, were subjects advised/educated on delivering at a health facility
 - **Response:** The pregnant women who participated in this trial were given messages at multiple time points during pregnancy advising them to deliver at health facilities. These educational messages were provided after the baseline data collection was completed where we ascertained the pregnant woman’s intended delivery location. Educational programs at antenatal clinic visits and during the two antenatal home visits by the ZamCAT field monitors (data collectors) during the course of pregnancy were used to encourage facility-based childbirth. This is described in the Methods section on page 3, lines 63-67.
- What were the primary outcome measures of ZamCAT?
 - **Response:** We have included the primary focus of the ZamCAT study and the primary outcomes in the Introduction (page 2, line 21-24). The primary outcome measures for ZamCAT, which were registered with ClinicalTrials.gov, were:
 - All-cause Neonatal Mortality [Time Frame: 28 days post-partum]
 - All-cause neonatal mortality based on vital status at 28 days post-partum
 - All-cause Neonatal Mortality Among Newborns Who Survived At least First Day of Life [Time Frame: 28 days post-partum]
 - All-cause mortality by day 28 of life among newborns who survive at least the first day of life
- First bullet under Strengths and Limitations section does not seem to have linkage with this article
 - **Response:** We have revised this first bullet to highlight the strength of the sample size on this observational analysis. Further, we have revised the strengths and limitations section in alignment with the recommendations from the editor (see below).
- Add citation for sentence on line 27-36 (*Given the emphasis on birth planning...*)
 - **Response:** Thank you for the recommendation. We have added the following citations to support the emphasis that has been made on birth planning or birth preparedness including a new systematic review:
 - McCauley H, Lowe K, Furtado N, Mangiaterra V, van den Broek N. What are the essential components of Antenatal Care? A systematic review of the literature and development of signal functions to guide monitoring and evaluation. BJOG. 2021 Nov 28. doi: 10.1111/1471-0528.17029. Epub ahead of print. PMID: 34839568. <https://pubmed.ncbi.nlm.nih.gov/34839568/>
 - Campbell OM, Graham WJ; Lancet Maternal Survival Series steering group. Strategies for reducing maternal mortality: getting on with what works. Lancet. 2006 Oct 7;368(9543):1284-99. doi: 10.1016/S0140-6736(06)69381-1. PMID: 17027735., <https://pubmed.ncbi.nlm.nih.gov/17027735/>
 - Bhutta ZA, Das JK, Bahl R, Lawn JE, Salam RA, Paul VK, et al. Can available interventions end preventable deaths in mothers, newborn babies, and stillbirths, and at what cost? Lancet. 2014;384(9940):347–70.
- Clarification of “secondary outcomes of ZamCAT study” under Methods
 - **Response:** The focus of this analysis was secondary outcomes measure #2 (highlighted) as found on clinicaltrials.gov for the ZamCAT study. We have included in this information in the Methods section (Page 2, Lines 45-49)
 - Secondary Outcome Measures:
 - Incidence of Omphalitis [Time Frame: 28 days postpartum]
 - Omphalitis, or umbilical cord infection, defined as:
 - presence of umbilical cord pus and mild, moderate or severe redness
 - moderate or severe redness without the presence of umbilical cord pus

- Place of Delivery [Time Frame: 28 days postpartum]
 - The location where mothers gave birth (home versus a health facility) will be compared to their planned delivery location.
 - Factors Influencing Delivery Location [Time Frame: 28 days postpartum]
 - Health facility characteristics and maternal decision-making factors that influence choice of delivery location (health facility vs. home delivery)
 - Health Facility Characteristics [Time Frame: 12 months after study initiation]
 - Characterization of the health services available to pregnant women, postpartum women and their offspring as assessed by comprehensive health facility and health worker surveys. This data was assessed and reported on 100 facilities (10 district hospitals and 90 health facilities).
- Totals of Table 1 and 2 do not match
 - **Response:** The tables 1 and 2 represent two different groups who were responsive to the questionnaires and we have added information about missing data where appropriate.
 - Table 1 has a total of 35,937 as those who responded to having “no plan” or “other” when asked about intended delivery location, and those who responded as “other” when asked about actual delivery location were excluded (n=1,168). This is included at the bottom of Table 1 with an asterisk.
 - Table 2 total is 37,105 which is consistent with the other tables. This is the number of the enrolled population for which intended and actual delivery information was available.
- Table 4: clarification on why need for skilled attendance and safety for mother/baby were chosen as the reason by those who intended to deliver at home and did.
 - **Response:** At the time of data collection at baseline and in the postpartum period, women were asked why they chose to deliver where they did. That list included eight potential answers, namely: Same location as prior delivery; Need for skilled attendance; Financial restraints; Distance; Relationship with Providers; Family/social expectations; Safety for mother/baby; or Other. As noted in our limitation sections, we do not have qualitative information to further understand why women chose the reason they did. Finally, as has been noted in the literature disrespectful care and abuse happens at the facility and thus, women could have chosen home birth as a matter of safety.
- In the discussion section, there was concern that results “could have been due to [the subjects] lack of comprehension about the question” about why they delivered at home as well as about “lack of comprehension about the term ‘skilled delivery.’”
 - **Response:** As noted above, we have only quantitative or categorical data on why women chose their intended and actual locations for delivery. It is possible that women perceived home birth as safer. We have added notes to the methods and discussion sections.
- Clarification on sentence “Although a policy change in Zambia resulted in TBAs no longer being trained and recommended that they stop conducting deliveries at home, a study on reasons for home delivery and use of TBAs in rural Zambia by Sialubanje et al. found that most women had positive attitudes towards TBAs and regarded them as respectful, skilled, and trustworthy.” Specifically, on why the ZamCAT trial included chlorhexidine application at home.
 - **Response:** The application of chlorhexidine was assessed at the facility and at home as a means to prevent neonatal mortality; that was the primary aim of the main trial. We have noted that goal in the introduction and in the methods; further, we have referenced the main outcomes paper where appropriate. However, this paper is focused on the secondary outcomes focused on birth location intention and practice. Therefore, we have maintained the focus of this paper on birth location.
- Clarification on sentence: “Moreover, most women indicated that negative attitudes toward the quality of services at the clinic as a reason for why they delivered at home.¹⁷” Specifically if the sentence is from reference #17 or from this study.

- **Response:** The sentence is captured from the results by Sialubanje et al. (reference #17) and is tied to the sentence directly before it which also references the same study.
- No reference to low status of women as a determinant in this study which could affect the intention versus actual practice in pregnant women as to where to deliver.
 - **Response:** Thank you for the point. *We don't have references for this statement and can't extrapolate from our data*
- Primary objectives of ZamCAT are not linked with the secondary objective that is the focus of this study.
 - **Response:** The data for this secondary analysis emerged from the primary study; thus, we feel it is important to note the main trials goals and how the data were collected.

VERSION 2 – REVIEW

REVIEWER	Singh, Amarjeet PGIMER, Department of Community Medicine
REVIEW RETURNED	26-Dec-2021
GENERAL COMMENTS	1. Please see the enclosed file. 2. A research is ethically required to advise the prospective mother about the IDEAL PLACE of delivery in such a study. It was not discussed. - The reviewer provided a marked copy with additional comments. Please contact the publisher for full details.

VERSION 2 – AUTHOR RESPONSE

Thank you for the opportunity to continue to strengthen this manuscript. We have copied the direct quotes from Reviewer #2 in the manuscript and responded to each query. We have included our responses and any edits made to the manuscript to the attached file.

Response to Reviewer Comments

Thank you for the opportunity to continue to strengthen this manuscript. We have copied the direct quotes from Reviewer #2 in the manuscript and responded to each query. We have included our responses and any edits made to the manuscript.

Reviewer 2:

- Comment: Abstract “Edit/reduce text”

O Response: The abstract fits the word count prescribed by the journal and we feel the details included are critical to the understanding of this pre-specified secondary analysis of a large cluster randomized controlled trial.

- Comment: Abstract/Secondary outcome measures “DECISION MAKING POWER??/in women”

O Response: The secondary outcome measures compared the mother's planned delivery location with the location where they actually gave birth (home vs. health facility). Our results encompass their decision-making around planned location and actual location of birth. We acknowledge that many factors may influence intent and practice when it comes to location of birth; thus, we have explored this further in the results and discussion of the paper.

• Comment: Abstract: Results "different period of gestation CAN NOT BE CLUBBED TOGETHER!"

O Response: As noted in the methods, we collected information on a woman's intended delivery location at the baseline/enrollment visit. The baseline data collection occurred in the antepartum period, typically during the second trimester. As we are reporting intended location as gathered at baseline, we have not changed the abstract.

• Comment: Abstract: Results "WHERE DID THE REST GO??" (in two places on this section)

O Response: As stated in the results section in the body of the manuscript, during the study 1,548 enrolled women (4% of the total) were excluded as a result of false pregnancies, miscarriages or abortion, withdrawal from study before delivery, lost to follow-up, or death. We had intended and actual delivery information on 37,105 of the enrolled population (97%). For the aggregate breakdown of where women intended to deliver and where women delivered, please see lines 118-122. As all results cannot fit in the abstract, we have retained the most important information in alignment with word count guidelines.

• Comment: Abstract: Conclusion: "IT WAS NOT A PART OF THE RCT or THE STUDY AS SUCH!"

O Response: This manuscript presents data from a pre-specified secondary analysis of the Zambia Chlorhexidine Application Trial (ZamCAT) (please see <https://clinicaltrials.gov/ct2/show/NCT01241318?term=ZamCAT&draw=2&rank=1> for verification); all women who enrolled shared their intended delivery location and associated reasons for that intention. Women were also followed to find out their actual practice and reasons why they delivered where they did. This secondary analysis does not utilize the randomization feature of the main trial (chlorhexidine or dry cord care), but rather answers important questions about why women intended and chose to deliver where they did. It would be unethical to randomize women to deliver in a facility or at home; thus, we use this rich dataset to answer important questions around motivations, intent, and actual practice. It is clear from the data that provision of birth planning and encouragement to deliver in the facility were insufficient to aid women in meeting their birth location intentions, which is why we have come to this conclusion in the abstract.

• Comment: Abstract: Conclusion: "(NOT STUDIED)"

O Response: In the abstract conclusion, we emphasize the main finding of the study—there were significant discrepancies between intended delivery location and actual delivery location. The birth planning and messaging provided to all women was insufficient to create concordance between intention and practice. We believe and call for more research to address barriers to achieving intention of a facility-based childbirth. We acknowledge that this was not studied in our analysis and are making a recommendation for future research as part of our conclusion.

• Comment: Body: Methods Patient and Public Involvement: Line 124-126: "ANY CHOICE RELATED DATA FROM WOMEN WHO WERE ENROLLED IN A TRIAL LIKE THIS, WILL BE BIASED...SINCE THEY MAY OPT MORE FOR AN HF DELIVERY!!"

O Response: We agree with the reviewer that study enrollment may have biased women's intentions and have noted that in our limitations section regarding social desirability bias (Discussion: lines 270-272). However, we would also note that the actual delivery location in the study cohort of >37,000 women in Southern Province, Zambia was similar to that of the Zambia DHS 2014 reported facility-

based childbirth coverage (56% facility-based childbirth); further supporting the representative nature of the population enrolled. (Citation: Central Statistical Office/Zambia, Ministry of Health/Zambia, University of Zambia Teaching Hospital Virology Laboratory, University of Zambia Department of Population Studies, Tropical Diseases Research Centre/Zambia, and ICF International. 2015. Zambia Demographic and Health Survey 2013-14. Rockville, Maryland, USA: Central Statistical Office/Zambia, Ministry of Health/Zambia, and ICF International. Available at <http://dhsprogram.com/pubs/pdf/FR304/FR304.pdf>.)

• Comment: Body: Results: Lines 143-144: “CAN AN 18-20 LADY DECIDE ALONE ABOUT THE INTENDED PLACE OF DELIVERY...NO DISCUSSION ON THE FAMILY DYNAMICS!!!??”

O Response: Yes, women of that age have the capacity, competency, ability, and legal right to decide where they intend to deliver. Pregnant women were asked about the reasons behind their choice for intended delivery location and actual location. As reported in Tables 3 and 4, family or societal expectations were identified as reasons by less than 10% of respondents. Therefore, we have focused our discussion on comparison of the key drivers of decision making and comparison to the literature. We have noted in the limitations (as highlighted in the response below) that the decision on where to deliver may have been made in consultation with family members or influenced by others (Discussion section, lines 272-277).

• Comment: “A research is ethically required to advise the prospective mother about the IDEAL PLACE of delivery in such a study. It was not discussed.”

O Response: As mentioned in the Methods section on page 3, lines 63-67, all pregnant women who participated in this trial were given messages at multiple time points during pregnancy advising them to deliver at health facilities. These educational messages were provided at the baseline visit, but after we gathered the enrollment data to ascertain the pregnant woman’s intended delivery location. Educational programs at antenatal clinic visits and during the two antenatal home visits by the ZamCAT field monitors (data collectors) were used to encourage facility-based childbirth. We have added in the revised manuscript that one limitation of our study design was that it would be impossible to rule out that in some cases the decision on where to deliver may have been made in consultation with the family unit or influenced by other forces outside the researchers’ control (Discussion section, lines 272-277). We understand the concerns of the reviewer, but again, the study encouraged enrolled women to deliver at a health facility and this message was repeated multiple times as part of the study visits.